# ACTION-DEPENDENT CONTROL VARIATES FOR POLICY OPTIMIZATION VIA STEIN'S IDENTITY

**Hao Liu**[*]
Computer Science
UESTC
Chengdu, China
uestcliuhao@gmail.com

**Yihao Feng**[*]
Computer science
University of Texas at Austin
Austin, TX, 78712
yihao@cs.utexas.edu

**Yi Mao**
Microsoft
Redmond, WA, 98052
maoyi@microsoft.com

**Dengyong Zhou**
Google
Kirkland, WA, 98033
dennyzhou@google.com

**Jian Peng**
Computer Science
UIUC
Urbana, IL 61801
jianpeng@illinois.edu

**Qiang Liu**
Computer Science
University of Texas at Austin
Austin, TX, 78712
lqiang@cs.utexas.edu

## ABSTRACT

Policy gradient methods have achieved remarkable successes in solving challenging reinforcement learning problems. However, it still often suffers from the large variance issue on policy gradient estimation, which leads to poor sample efficiency during training. In this work, we propose a control variate method to effectively reduce variance for policy gradient methods. Motivated by the Stein's identity, our method extends the previous control variate methods used in REINFORCE and advantage actor-critic by introducing more general action-dependent baseline functions. Empirical studies show that our method significantly improves the sample efficiency of the state-of-the-art policy gradient approaches.

## 1 INTRODUCTION

Deep reinforcement learning (RL) provides a general framework for solving challenging goal-oriented sequential decision-making problems, It has recently achieved remarkable successes in advancing the frontier of AI technologies (Silver et al., 2017; Mnih et al., 2013; Silver et al., 2016; Schulman et al., 2017). Policy gradient (PG) is one of the most successful model-free RL approaches that has been widely applied to high dimensional continuous control, vision-based navigation and video games (Schulman et al., 2016; Kakade, 2002; Schulman et al., 2015; Mnih et al., 2016).

Despite these successes, a key problem of policy gradient methods is that the gradient estimates often have high variance. A naive solution to fix this issue would be generating a large amount of rollout samples to obtain a reliable gradient estimation in each step. Regardless of the cost of generating large samples, in many practical applications like developing driverless cars, it may not even be possible to generate as many samples as we want. A variety of variance reduction techniques have been proposed for policy gradient methods (See e.g. Weaver & Tao 2001, Greensmith et al. 2004, Schulman et al. 2016 and Asadi et al. 2017).

In this work, we focus on the control variate method, one of the most widely used variance reduction techniques in policy gradient and variational inference. The idea of the control variate method is to subtract a Monte Carlo gradient estimator by a baseline function that analytically has zero expectation. The resulted estimator does not introduction biases theoretically, but may achieve much lower variance if the baseline function is properly chosen such that it cancels out the variance of the original gradient estimator. Different control variates yield different variance reduction methods. For example, in REINFORCE (Williams, 1992), a constant baseline function is chosen as a control variate; advantage actor-critic (A2C) (Sutton & Barto, 1998; Mnih et al., 2016) considers a state-dependent baseline function as the control variate, which is often set to be an estimated value

---

[*]Both authors contributed equally. Author ordering determined by coin flip over a Google Hangout.

function $V(s)$. More recently, in Q-prop (Gu et al., 2016b), a more general baseline function that linearly depends on the actions is proposed and shows promising results on several challenging tasks. It is natural to expect even more flexible baseline functions which depend on both states and actions to yield powerful variance reduction. However, constructing such baseline functions turns out to be fairly challenging, because it requires new and more flexible mathematical identities that can yield a larger class of baseline functions with zero analytic expectation under the policy distribution of interest.

To tackle this problem, we sort to the so-called *Stein's identity* (Stein, 1986) which defines a broad class of identities that are sufficient to fully characterize the distribution under consideration (see e.g., Liu et al., 2016; Chwialkowski et al., 2016). By applying the Stein's identity and also drawing connection with the reparameterization trick (Kingma & Welling, 2013; Rezende et al., 2014), we construct a class of *Stein control variate* that allows us to use arbitrary baseline functions that depend on both actions and states. Our approach tremendously extends the existing control variates used in REINFORCE, A2C and Q-prop.

We evaluate our method on a variety of reinforcement learning tasks. Our experiments show that our Stein control variate can significantly reduce the variance of gradient estimation with more flexible and nonlinear baseline functions. When combined with different policy optimization methods, including both proximal policy optimization (PPO) (Schulman et al., 2017; Heess et al., 2017) and trust region policy optimization (TRPO) (Schulman et al., 2015; 2016), it greatly improves the sample efficiency of the entire policy optimization.

## 2 BACKGROUND

We first introduce basic backgrounds of reinforcement learning and policy gradient and set up the notation that we use in the rest of the paper in Section 2.1, and then discuss the control variate method as well as its application in policy gradient in Section 2.2.

### 2.1 REINFORCEMENT LEARNING AND POLICY GRADIENT

Reinforcement learning considers the problem of finding an optimal policy for an agent which interacts with an uncertain environment and collects reward per action. The goal of the agent is to maximize the long-term cumulative reward. Formally, this problem can be formulated as a Markov decision process over the environment states $s \in S$ and agent actions $a \in A$, under an unknown environmental dynamic defined by a transition probability $T(s'|s, a)$ and a reward signal $r(s, a)$ immediately following the action $a$ performed at state $s$. The agent's action $a$ is selected by a conditional probability distribution $\pi(a|s)$ called policy. In policy gradient methods, we consider a set of candidate policies $\pi_\theta(a|s)$ parameterized by $\theta$ and obtain the optimal policy by maximizing the expected cumulative reward or return

$$J(\theta) = \mathbb{E}_{s \sim \rho_\pi, a \sim \pi(a|s)} \left[ r(s, a) \right],$$

where $\rho_\pi(s) = \sum_{t=1}^{\infty} \gamma^{t-1} \Pr(s_t = s)$ is the normalized discounted state visitation distribution with discount factor $\gamma \in [0, 1)$. To simplify the notation, we denote $\mathbb{E}_{s \sim \rho_\pi, a \sim \pi(a|s)}[\cdot]$ by simply $\mathbb{E}_\pi[\cdot]$ in the rest of paper. According to the policy gradient theorem (Sutton & Barto, 1998), the gradient of $J(\theta)$ can be written as

$$\nabla_\theta J(\theta) = \mathbb{E}_\pi \left[ \nabla_\theta \log \pi(a|s) Q^\pi(s, a) \right], \tag{1}$$

where $Q^\pi(s, a) = \mathbb{E}_\pi \left[ \sum_{t=1}^{\infty} \gamma^{t-1} r(s_t, a_t) | s_1 = s, a_1 = a \right]$ denotes the expected return under policy $\pi$ starting from state $s$ and action $a$. Different policy gradient methods are based on different stochastic estimation of the expected gradient in Eq (1). Perhaps the most straightforward way is to simulate the environment with the current policy $\pi$ to obtain a trajectory $\{(s_t, a_t, r_t)\}_{t=1}^n$ and estimate $\nabla_\theta J(\theta)$ using the Monte Carlo estimation:

$$\hat{\nabla}_\theta J(\theta) = \frac{1}{n} \sum_{t=1}^{n} \gamma^{t-1} \nabla_\theta \log \pi(a_t|s_t) \hat{Q}^\pi(s_t, a_t), \tag{2}$$

where $\hat{Q}^\pi(s_t, a_t)$ is an empirical estimate of $Q^\pi(s_t, a_t)$, e.g., $\hat{Q}^\pi(s_t, a_t) = \sum_{j \geq t} \gamma^{j-t} r_j$. Unfortunately, this naive method often introduces large variance in gradient estimation. It is almost always

the case that we need to use control variates method for variance reduction, which we will introduce in the following. It has been found that biased estimators help improve the performance, e.g., by using biased estimators of $Q^\pi$ or dropping the $\gamma^{t-1}$ term in Eq (2). In this work, we are interested in improving the performance without introducing additional biases, at least theoretically.

## 2.2 CONTROL VARIATE

The control variates method is one of the most widely used variance reduction techniques in policy gradient. Suppose that we want to estimate the expectation $\mu = \mathbb{E}_\tau[g(s,a)]$ with Monte Carlo samples $(s_t, a_t)_{t=1}^n$ drawn from some distribution $\tau$, which is assumed to have a large variance $\mathrm{var}_\tau(g)$. The *control variate* is a function $f(s,a)$ with known analytic expectation under $\tau$, which, without losing of generality, can be assumed to be zero: $\mathbb{E}_\tau[f(s,a)] = 0$. With $f$, we can have an alternative unbiased estimator

$$\hat{\mu} = \frac{1}{n} \sum_{t=1}^n \left( g(s_t, a_t) - f(s_t, a_t) \right),$$

where the variance of this estimator is $\mathrm{var}_\tau(g - f)/n$, instead of $\mathrm{var}_\tau(g)/n$ for the Monte Carlo estimator. By taking $f$ to be similar to $g$, e.g. $f = g - \mu$ in the ideal case, the variance of $g - f$ can be significantly reduced, thus resulting in a more reliable estimator.

The key step here is to find an identity that yields a large class of functional $f$ with zero expectation. In most existing policy gradient methods, the following identity is used

$$\mathbb{E}_{\pi(a|s)}\left[\nabla_\theta \log \pi(a|s)\phi(s)\right] = 0, \quad \text{for any function } \phi. \tag{3}$$

Combining it with the policy gradient theorem, we obtain

$$\hat{\nabla}_\theta J(\theta) = \frac{1}{n} \sum_{t=1}^n \nabla_\theta \log \pi(a_t|s_t) \left( \hat{Q}^\pi(s_t, a_t) - \phi(s_t) \right), \tag{4}$$

Note that we drop the $\gamma^{t-1}$ term in Eq (2) as we do in practice. The introduction of the function $\phi$ does not change the expectation but can decrease the variance significantly when it is chosen properly to cancel out the variance of $Q^\pi(s,a)$. In REINFORCE, $\phi$ is set to be a constant $\phi(s) = b$ baseline, and $b$ is usually set to approximate the average reward, or determined by minimizing $\mathrm{var}(\hat{\nabla}_\theta J(\theta))$ empirically. In advantage actor-critic (A2C), $\phi(s)$ is set to be an estimator of the value function $V^\pi(s) = \mathbb{E}_{\pi(a|s)}[Q^\pi(s,a)]$, so that $\hat{Q}^\pi(s,a) - \phi(s)$ is an estimator of the advantage function. For notational consistency, we call $\phi$ the baseline function and $f(s,a) = \nabla_\theta \log \pi(a|s)\phi(s)$ the corresponding control variate.

Although REINFORCE and A2C have been widely used, their applicability is limited by the possible choice of $\phi$. Ideally, we want to set $\phi$ to equal $Q^\pi(s,a)$ up to a constant to reduce the variance of $\hat{\nabla}_\theta J(\theta)$ to close to zero. However, this is impossible for REINFORCE or A2C because $\phi(s)$ only depends on state $s$ but not action $a$ by its construction. Our goal is to develop a more general control variate that yields much smaller variance of gradient estimation than the one in Eq (4).

## 3 POLICY GRADIENT WITH STEIN CONTROL VARIATE

In this section, we present our Stein control variate for policy gradient methods. We start by introducing Stein's identity in Section 3.1, then develop in Section 3.2 a variant that yields a new control variate for policy gradient and discuss its connection to the reparameterization trick and the Q-prop method. We provide approaches to estimate the optimal baseline functions in Section 3.3, and discuss the special case of the Stein control variate for Gaussian policies in Section 3.4. We apply our control variate to proximal policy optimization (PPO) in Section 3.5.

### 3.1 STEIN'S IDENTITY

Given a policy $\pi(a|s)$, Stein's identity w.r.t $\pi$ is

$$\mathbb{E}_{\pi(a|s)}\left[\nabla_a \log \pi(a|s)\phi(s,a) + \nabla_a\phi(s,a)\right] = 0, \quad \forall s, \tag{5}$$

which holds for any real-valued function $\phi(s, a)$ with some proper conditions. To see this, note the left hand side of Eq (5) is equivalent to $\int \nabla_a (\pi(a|s)\phi(s, a)) \, da$, which equals zero if $\pi(a|s)\phi(s, a)$ equals zero on the boundary of the integral domain, or decay sufficiently fast (e.g., exponentially) when the integral domain is unbounded.

The power of Stein's identity lies in the fact that it defines an infinite set of identities, indexed by arbitrary function $\phi(s, a)$, which is sufficient to uniquely identify a distribution as shown in the work of Stein's method for proving central limit theorems (Stein, 1986; Barbour & Chen, 2005), goodness-of-fit test (Chwialkowski et al., 2016; Liu et al., 2016), and approximate inference (Liu & Wang, 2016). Oates et al. (2017) has applied Stein's identity as a control variate for general Monte Carlo estimation, which is shown to yield a *zero-variance* estimator because the control variate is flexible enough to approximate the function of interest arbitrarily well.

## 3.2 STEIN CONTROL VARIATE FOR POLICY GRADIENT

Unfortunately, for the particular case of policy gradient, it is not straightforward to directly apply Stein's identity (5) as a control variate, since the dimension of the left-hand side of (5) does not match the dimension of a policy gradient: the gradient in (5) is taken w.r.t. the action $a$, while the policy gradient in (1) is taken w.r.t. the parameter $\theta$. Therefore, we need a general approach to connect $\nabla_a \log \pi(a|s)$ to $\nabla_\theta \log \pi(a|s)$ in order to apply Stein's identity as a control variate for policy gradient. We show in the following theorem that this is possible when the policy is *reparameterizable* in that $a \sim \pi_\theta(a|s)$ can be viewed as generated by $a = f_\theta(s, \xi)$ where $\xi$ is a random noise drawn from some distribution independently of $\theta$. With an abuse of notation, we denote by $\pi(a, \xi|s)$ the joint distribution of $(a, \xi)$ conditioned on $s$, so that $\pi(a|s) = \int \pi(a|s, \xi)\pi(\xi)d\xi$, where $\pi(\xi)$ denotes the distribution generating $\xi$ and $\pi(a|s, \xi) = \delta(a - f(s, \xi))$ where $\delta$ is the Delta function.

**Theorem 3.1.** *With the reparameterizable policy defined above, using Stein's identity, we can derive*

$$\mathbb{E}_{\pi(a|s)} \left[ \nabla_\theta \log \pi(a|s)\phi(s, a) \right] = \mathbb{E}_{\pi(a, \xi|s)} \left[ \nabla_\theta f_\theta(s, \xi)\nabla_a \phi(s, a) \right]. \tag{6}$$

*Proof.* See Appendix for the detail proof. To help understand the intuition, we can consider the Delta function as a Gaussian with a small variance $h^2$, i.e. $\pi(a|s, \xi) \propto \exp(-\|a - f(s, \xi)\|_2^2/2h^2)$, for which it is easy to show that

$$\nabla_\theta \log \pi(a, \xi \mid s) = -\nabla_\theta f_\theta(s, \xi) \, \nabla_a \log \pi(a, \xi \mid s). \tag{7}$$

This allows us to convert between the derivative w.r.t. $a$ and w.r.t. $\theta$, and apply Stein's identity. $\quad\square$

**Stein Control Variate**    Using Eq (6) as a control variate, we obtain the following general formula of policy gradient:

$$\nabla_\theta J(\theta) = \mathbb{E}_\pi \left[ \nabla_\theta \log \pi(a|s)(Q^\pi(s, a) - \phi(s, a)) \, + \, \nabla_\theta f_\theta(s, \xi)\nabla_a \phi(s, a) \right], \tag{8}$$

where any fixed choice of $\phi$ does not introduce bias to the expectation. Given a sample set $(s_t, a_t, \xi_t)_{t=1}^n$ where $a_t = f_\theta(s_t, \xi_t)$, an estimator of the gradient is

$$\hat{\nabla}_\theta J(\theta) = \frac{1}{n} \sum_{t=1}^n \left[ \nabla_\theta \log \pi(a_t \mid s_t)(\hat{Q}^\pi(s_t, a_t) - \phi(s_t, a_t)) + \nabla_\theta f_\theta(s_t, \xi_t)\nabla_a \phi(s_t, a_t) \right]. \tag{9}$$

This estimator clearly generalizes the control variates used in A2C and REINFORCE. To see this, let $\phi$ be action independent, i.e. $\phi(s, a) = \phi(s)$ or even $\phi(s, a) = b$, in both cases the last term in (9) equals to zero because $\nabla_a \phi = 0$. When $\phi$ is action-dependent, the last term (9) does not vanish in general, and in fact will play an important role for variance reduction as we will illustrate later.

**Relation to Q-prop**    Q-prop is a recently introduced sample-efficient policy gradient method that constructs a general control variate using Taylor expansion. Here we show that Q-prop can be derived from (8) with a special $\phi$ that depends on the action linearly, so that its gradient w.r.t. $a$ is action-independent, i.e. $\nabla_a \phi(a, s) = \varphi(s)$. In this case, Eq (8) becomes

$$\nabla_\theta J(\theta) = \mathbb{E}_\pi \left[ \nabla_\theta \log \pi(a|s)(Q^\pi(s, a) - \phi(s, a)) \, + \, \nabla_\theta f_\theta(s, \xi)\varphi(s) \right].$$

Furthermore, note that $\mathbb{E}_{\pi(\xi)}[\nabla_\theta f(s, \xi)] = \nabla_\theta \mathbb{E}_{\pi(\xi)}[f(s, \xi)] := \nabla_\theta \mu_\pi(s)$, where $\mu_\pi(s)$ is the expectation of the action conditioned on $s$. Therefore,

$$\nabla_\theta J(\theta) = \mathbb{E}_\pi \left[ \nabla_\theta \log \pi(a|s) \left( Q^\pi(s, a) - \phi(s, a) \right) + \nabla_\theta \mu_\pi(s) \varphi(s) \right],$$

which is the identity used in Q-prop to construct their control variate (see Eq 6 in Gu et al. 2016b). In Q-prop, the baseline function is constructed empirically by the first-order Taylor expansion as

$$\phi(s, a) = \hat{V}^\pi(s) + \langle \nabla_a \hat{Q}^\pi(s, \mu_\pi(s)), \ a - \mu_\pi(s) \rangle, \tag{10}$$

where $\hat{V}^\pi(s)$ and $\hat{Q}^\pi(s, a)$ are parametric functions that approximate the value function and Q function under policy $\pi$, respectively. In contrast, our method allows us to use more general and flexible, nonlinear baseline functions $\phi$ to construct the Stein control variate which is able to decrease the variance more significantly.

**Relation to the reparameterization trick** The identity (6) is closely connected to the reparameterization trick for gradient estimation which has been widely used in variational inference recently (Kingma & Welling, 2013; Rezende et al., 2014). Specifically, let us consider an auxiliary objective function based on function $\phi$:

$$L_s(\theta) := \mathbb{E}_{\pi(a|s)}[\phi(s, a)] = \int \pi(a|s)\phi(s, a)da.$$

Then by the log-derivative trick, we can obtain the gradient of this objective function as

$$\nabla_\theta L_s(\theta) = \int \nabla_\theta \pi(a|s)\phi(s, a)da = \mathbb{E}_{\pi(a|s)} \left[ \nabla_\theta \log \pi(a|s)\phi(s, a) \right], \tag{11}$$

which is the left-hand side of (6). On the other hand, if $a \sim \pi(a|s)$ can be parameterized by $a = f_\theta(s, \xi)$, then $L_s(\theta) = \mathbb{E}_{\pi(\xi)}[\phi(s, f_\theta(s, \xi))]$, leading to the reparameterized gradient in Kingma & Welling (2013):

$$\nabla_\theta L_s(\theta) = \mathbb{E}_{\pi(a, \xi|s)} \left[ \nabla_\theta f_\theta(s, \xi)\nabla_a \phi(s, a) \right]. \tag{12}$$

Equation (11) and (12) are equal to each other since both are $\nabla_\theta L_s(\theta)$. This provides another way to prove the identity in (6). The connection between Stein's identity and the reparameterization trick that we reveal here is itself interesting, especially given that both of these two methods have been widely used in different areas.

## 3.3 CONSTRUCTING THE BASELINE FUNCTIONS FOR STEIN CONTROL VARIATE

We need to develop practical approaches to choose the baseline functions $\phi$ in order to fully leverage the power of the flexible Stein control variate. In practice, we assume a flexible parametric form $\phi_w(s, a)$ with parameter $w$, e.g. linear functions or neural networks, and hope to optimize $w$ efficiently for variance reduction. Here we introduce two approaches for optimizing $w$ and discuss some practical considerations.

We should remark that if $\phi$ is constructed based on data $(s_t, a_t)_{t=1}^n$, it introduces additional dependency and (4) is no longer an unbiased estimator theoretically. However, the bias introduced this way is often negligible in practice (see e.g., Section 2.3.4 of Oates & Girolami (2016)). For policy gradient, this bias can be avoided by estimating $\phi$ based on the data from the previous iteration.

**Estimating $\phi$ by Fitting Q Function** Eq (8) provides an interpolation between the log-likelihood ratio policy gradient (1) (by taking $\phi = 0$) and a reparameterized policy gradient as follows (by taking $\phi(s, a) = Q^\pi(s, a)$):

$$\nabla_\theta J(\theta) = \mathbb{E}_\pi[\nabla_\theta f(s, \xi)\nabla_a Q^\pi(s, a)]. \tag{13}$$

It is well known that the reparameterized gradient tends to yield much smaller variance than the log-likelihood ratio gradient from the variational inference literature (see e.g., Kingma & Welling, 2013; Rezende et al., 2014; Roeder et al., 2017; Tucker et al., 2017). An intuitive way to see this is to consider the extreme case when the policy is deterministic. In this case the variance of (11) is infinite because $\log \pi(a|s)$ is either infinite or does not exist, while the variance of (12) is zero because $\xi$ is

deterministic. Because optimal policies often tend to be close to deterministic, the reparameterized gradient should be favored for smaller variance.

Therefore, one natural approach is to set $\phi$ to be close to Q function, that is, $\phi(s, a) = \hat{Q}^\pi(s, a)$ so that the log-likelihood ratio term is small. Any methods for Q function estimation can be used. In our experiments, we optimize the parameter $w$ in $\phi_w(s, a)$ by

$$\min_w \sum_{t=1}^n (\phi_w(s_t, a_t) - R_t)^2, \tag{14}$$

where $R_t$ an estimate of the reward starting from $(s_t, a_t)$. It is worth noticing that with deterministic policies, (13) is simplified to the update of deep deterministic policy gradient (DDPG) (Lillicrap et al., 2015; Silver et al., 2014). However, DDPG directly plugs an estimator $\hat{Q}^\pi(s, a)$ into (13) to estimate the gradient, which may introduce a large bias; our formula (8) can be viewed as correcting this bias in the reparameterized gradient using the log-likelihood ratio term.

**Estimating $\phi$ by Minimizing the Variance**   Another approach for obtaining $\phi$ is to directly minimize the variance of the gradient estimator. Note that $\text{var}(\hat{\nabla}_\theta J(\theta)) = \mathbb{E}[(\hat{\nabla}_\theta J(\theta))^2] - \mathbb{E}[\hat{\nabla}_\theta J(\theta)]^2$. Since $\mathbb{E}[\hat{\nabla}_\theta J(\theta)] = \nabla_\theta J(\theta)$ which does not depend on $\phi$, it is sufficient to minimize the first term. Specifically, for $\phi_w(s, a)$ we optimize $w$ by

$$\min_w \sum_{t=1}^n \left\| \nabla_\theta \log \pi(a_t \mid s_t) \left( \hat{Q}^\pi(s_t, a_t) - \phi_w(s_t, a_t) \right) + \nabla_\theta f(s_t, \xi_t) \nabla_a \phi_w(s_t, a_t) \right\|_2^2. \tag{15}$$

In practice, we find that it is difficult to implement this efficiently using the auto-differentiation in the current deep learning platforms because it involves derivatives w.r.t. both $\theta$ and $a$. We develop a computational efficient approximation for the special case of Gaussian policy in Section 3.4.

**Architectures of $\phi$**   Given the similarity between $\phi$ and the Q function as we mentioned above, we may decompose $\phi$ into

$$\phi_w(s, a) = \hat{V}^\pi(s) + \psi_w(s, a).$$

The term $\hat{V}^\pi(s)$ is parametric function approximation of the value function which we separately estimate in the same way as in A2C, and $w$ is optimized using the method above with fixed $\hat{V}^\pi(s)$. Here the function $\psi_w(s, a)$ can be viewed as an estimate of the advantage function, whose parameter $w$ is optimized using the two optimization methods introduced above. To see, we rewrite our gradient estimator to be

$$\hat{\nabla}_\theta J(\theta) = \frac{1}{n} \sum_{t=1}^n \left[ \nabla_\theta \log \pi(a_t \mid s_t)(\hat{A}^\pi(s_t, a_t) - \psi_w(s_t, a_t)) + \nabla_\theta f_\theta(s_t, \xi_t) \nabla_a \psi_w(s_t, a_t) \right], \tag{16}$$

where $\hat{A}^\pi(s_t, a_t) = \hat{Q}^\pi(s_t, a_t) - \hat{V}^\pi(s_t)$ is an estimator of the advantage function. If we set $\psi_w(s, a) = 0$, then Eq (16) clearly reduces to A2C. We find that separating $\hat{V}^\pi(s)$ from $\psi_w(s, a)$ works well in practice, because it effectively provides a useful initial estimation of $\phi$, and allows us to directly improve the $\phi$ on top of the value function baseline.

### 3.4   STEIN CONTROL VARIATE FOR GAUSSIAN POLICIES

Gaussian policies have been widely used and are shown to perform efficiently in many practical continuous reinforcement learning settings. Because of their wide applicability, we derive the gradient estimator with the Stein control variate for Gaussian policies here and use it in our experiments.

Specifically, Gaussian policies take the form $\pi(a \mid s) = \mathcal{N}(a; \mu_{\theta_1}(s), \Sigma_{\theta_2}(s))$, where mean $\mu$ and covariance matrix $\Sigma$ are often assumed to be parametric functions with parameters $\theta = [\theta_1, \theta_2]$. This is equivalent to generating $a$ by $a = f_\theta(s, \xi) = \mu_{\theta_1}(s) + \Sigma_{\theta_2}(s)^{1/2}\xi$, where $\xi \sim \mathcal{N}(0, 1)$. Following Eq (8), the policy gradient w.r.t. the mean parameter $\theta_1$ is

$$\nabla_{\theta_1} J(\theta) = \mathbb{E}_\pi \left[ \nabla_{\theta_1} \log \pi(a|s) \left( Q^\pi(s, a) - \phi(s, a) \right) + \nabla_{\theta_1} \mu(s) \nabla_a \phi(s, a) \right]. \tag{17}$$

For each coordinate $\theta_\ell$ in the variance parameter $\theta_2$, its gradient is computed as

$$\nabla_{\theta_\ell} J(\theta) = \mathbb{E}_\pi \left[ \nabla_{\theta_\ell} \log \pi(a|s) \left( Q^\pi(s,a) - \phi(s,a) \right) - \frac{1}{2} \left\langle \nabla_a \log \pi(a|s) \nabla_a \phi(s,a)^\top, \nabla_{\theta_\ell} \Sigma \right\rangle \right], \quad (18)$$

where $\langle A, \ B \rangle := \mathrm{trace}(AB)$ for two $d_a \times d_a$ matrices.

Note that the second term in (18) contains $\nabla_a \log \pi(a|s)$; we can further apply Stein's identity on it to obtain a simplified formula

$$\nabla_{\theta_\ell} J(\theta) = \mathbb{E}_\pi \left[ \nabla_{\theta_\ell} \log \pi(a|s) \left( Q^\pi(s,a) - \phi(s,a) \right) \ + \ \frac{1}{2} \left\langle \nabla_{a,a} \phi(s,a), \ \nabla_{\theta_\ell} \Sigma \right\rangle \right]. \quad (19)$$

The estimator in (19) requires to evaluate the second order derivative $\nabla_{a,a}\phi$, but may have lower variance compared to that in (18). To see this, note that if $\phi(s,a)$ is a linear function of $a$ (like the case of Q-prop), then the second term in (19) vanishes to zero, while that in (18) does not.

We also find it is practically convenient to estimate the parameters $w$ in $\phi$ by minimizing $\mathrm{var}(\hat{\nabla}_\mu J) + \mathrm{var}(\hat{\nabla}_\Sigma J)$, instead of the exact variance $\mathrm{var}(\hat{\nabla}_\theta J)$. Further details can be found in Appendix 7.2.

## 3.5 PPO with Stein control variate

Proximal Policy Optimization (PPO) (Schulman et al., 2017; Heess et al., 2017) is recently introduced for policy optimization. It uses a proximal Kullback-Leibler (KL) divergence penalty to regularize and stabilize the policy gradient update. Given an existing policy $\pi_{\mathrm{old}}$, PPO obtains a new policy by maximizing the following surrogate loss function

$$J_{\mathrm{ppo}}(\theta) = \mathbb{E}_{\pi_{\mathrm{old}}} \left[ \frac{\pi_\theta(a|s)}{\pi_{\mathrm{old}}(a|s)} Q^\pi(s,a) - \lambda \mathrm{KL} \left[ \pi_{\mathrm{old}}(\cdot|s) \ || \ \pi_\theta(\cdot|s) \right] \right],$$

where the first term is an approximation of the expected reward, and the second term enforces the the updated policy to be close to the previous policy under KL divergence. The gradient of $J_{\mathrm{ppo}}(\theta)$ can be rewritten as

$$\nabla_\theta J_{\mathrm{ppo}}(\theta) = \mathbb{E}_{\pi_{\mathrm{old}}} \left[ w_\pi(s,a) \nabla_\theta \log \pi(a|s) Q^\pi_\lambda(s,a) \right]$$

where $w_\pi(s,a) := \pi_\theta(a|s)/\pi_{\mathrm{old}}(a|s)$ is the density ratio of the two polices, and $Q^\pi_\lambda(s,a) := Q^\pi(s,a) + \lambda w_\pi(s,a)^{-1}$ where the second term comes from the KL penalty. Note that $\mathbb{E}_{\pi_{\mathrm{old}}}[w_\pi(s,a) f(s,a)] = \mathbb{E}_\pi[f(s,a)]$ by canceling the density ratio. Applying (6), we obtain

$$\nabla_\theta J_{\mathrm{ppo}}(\theta) = \mathbb{E}_{\pi_{\mathrm{old}}} \left[ w_\pi(s,a) \left( \nabla_\theta \log \pi(a|s) \left( Q^\pi_\lambda(s,a) - \phi(s,a) \right) \ + \ \nabla_\theta f_\theta(s,a) \nabla_a \phi(s,a) \right) \right]. \quad (20)$$

Putting everything together, we summarize our PPO algorithm with Stein control variates in Algorithm 1. It is also straightforward to integrate the Stein control variate with TRPO.

## 4 Related Work

Stein's identity has been shown to be a powerful tool in many areas of statistical learning and inference. An incomplete list includes Gorham & Mackey (2015), Oates et al. (2017), Oates et al. (2016), Chwialkowski et al. (2016), Liu et al. (2016), Sedghi et al. (2016), Liu & Wang (2016), Feng et al. (2017), Liu & Lee (2017). This work was originally motivated by Oates et al. (2017), which uses Stein's identity as a control variate for general Monte Carlo estimation. However, as discussed in Section 3.1, the original formulation of Stein's identity can not be directly applied to policy gradient, and we need the mechanism introduced in (6) that also connects to the reparameterization trick (Kingma & Welling, 2013; Rezende et al., 2014).

Control variate method is one of the most widely used variance reduction techniques in policy gradient (see e.g., Greensmith et al., 2004). However, action-dependent baselines have not yet been well

---

**Algorithm 1** PPO with Control Variate through Stein's Identity (the PPO procedure is adapted from Algorithm 1 in Heess et al. 2017)

---

**repeat**
    Run policy $\pi_\theta$ for $n$ timesteps, collecting $\{s_t, a_t, \xi_t, r_t\}$, where $\xi_t$ is the random seed that generates action $a_t$, i.e., $a_t = f_\theta(s_t, \xi_t)$. Set $\pi_{\text{old}} \leftarrow \pi_\theta$.

    *// Updating the baseline function $\phi$*
    **for** K iterations **do**
        Update $w$ by one stochastic gradient descent step according to (14), or (15), or (27) for Gaussian policies.
    **end for**

    *// Updating the policy $\pi$*
    **for** M iterations **do**
        Update $\theta$ by one stochastic gradient descent step with (20) (adapting it with (17) and (19) for Gaussian policies).
    **end for**

    *// Adjust the KL penalty coefficient $\lambda$*
    **if** $\text{KL}[\pi_{\text{old}}|\pi_\theta] > \beta_{\text{high}}\text{KL}_{\text{target}}$ **then**
        $\lambda \leftarrow \alpha\lambda$
    **else if** $\text{KL}[\pi_{\text{old}}|\pi_\theta] < \beta_{\text{low}}\text{KL}_{\text{target}}$ **then**
        $\lambda \leftarrow \lambda/\alpha$
    **end if**
**until** Convergence

---

studied. Besides Q-prop (Gu et al., 2016b) which we draw close connection to, the work of Thomas & Brunskill (2017) also suggests a way to incorporate action-dependent baselines, but is restricted to the case of compatible function approximation. More recently, Tucker et al. (2017) studied a related action-dependent control variate for discrete variables in learning latent variable models.

Recently, Gu et al. (2017) proposed an interpolated policy gradient (IPG) framework for integrating on-policy and off-policy estimates that generalizes various algorithms including DDPG and Q-prop. If we set $\nu = 1$ and $p^\pi = p^\beta$ (corresponding to using off-policy data purely) in IPG, it reduces to a special case of (16) with $\psi_w(s, a) = Q_w(s, a) - \mathbb{E}_{\pi(a|s)}[Q_w(s, a)]$ where $Q_w$ an approximation of the Q-function. However, the emphasis of Gu et al. (2017) is on integrating on-policy and off-policy estimates, generally yielding theoretical bias, and the results of the case when $\nu = 1$ and $p^\pi = p^\beta$ were not reported. Our work presents the result that shows significant improvement of sample efficiency in policy gradient by using nonlinear, action-dependent control variates.

In parallel to our work, there have been some other works discovered action-dependent baselines for policy-gradient methods in reinforcement learning. Such works include Grathwohl et al. (2018) which train an action-dependent baseline for both discrete and continuous control tasks. Wu et al. (2018) exploit per-dimension independence of the action distribution to produce an action-dependent baseline in continuous control tasks.

## 5 EXPERIMENTS

We evaluated our control variate method when combining with PPO and TRPO on continuous control environments from the OpenAI Gym benchmark (Brockman et al., 2016) using the MuJoCo physics simulator (Todorov et al., 2012). We show that by using our more flexible baseline functions, we can significantly improve the sample efficiency compared with methods based on the typical value function baseline and Q-prop.

All our experiments use Gaussian policies. As suggested in Section 3.3, we assume the baseline to have a form of $\phi_w(s, a) = \hat{V}^\pi(s) + \psi_w(s, a)$, where $\hat{V}^\pi$ is the valued function estimated separately in the same way as the value function baseline, and $\psi_w(s, a)$ is a parametric function whose value $w$ is decided by minimizing either Eq (14) (denoted by `FitQ`), or Eq (27) designed for Gaussian policy (denoted by `MinVar`). We tested three different architectures of $\psi_w(s, a)$, including

`Linear.` $\psi_w(s, a) = \langle \nabla_a q_w(a, \mu_\pi(s)), (a - \mu_\pi(s)) \rangle$, where $q_w$ is a parametric function designed for estimating the Q function $Q^\pi$. This structure is motivated by Q-prop, which estimates $w$ by

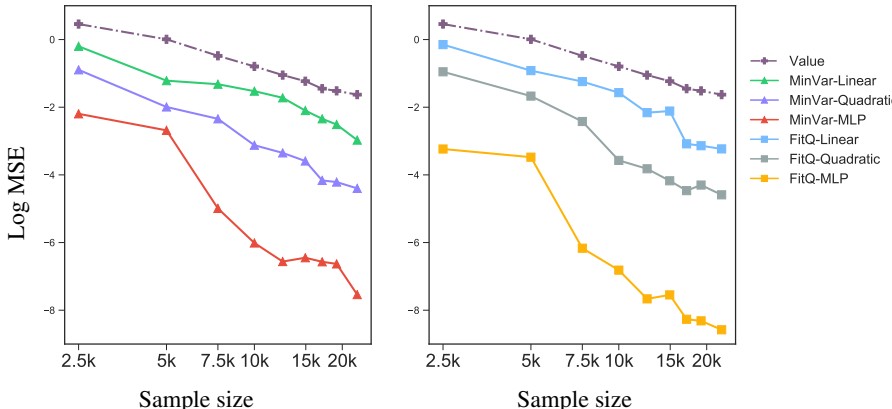

Figure 1: The variance of gradient estimators of different control variates under a fixed policy obtained by running vanilla PPO for 200 iterations in the Walker2d-v1 environment.

fitting $q_w(s, a)$ with $Q^\pi$. Our `MinVar`, and `FitQ` methods are different in that they optimize $w$ as a part of $\phi_w(s, a)$ by minimizing the objective in Eq (14) and Eq (27). We show in Section 5.2 that our optimization methods yield better performance than Q-prop even with the same architecture of $\psi_w$. This is because our methods directly optimize for the baseline function $\phi_w(s, a)$, instead of $q_w(s, a)$ which serves an intermediate step.

`Quadratic`. $\psi_w(s, a) = -(a - \mu_w(s))^\top \Sigma_w^{-1}(a - \mu_w(s))$. In our experiments, we set $\mu_w(s)$ to be a neural network, and $\Sigma_w$ a positive diagonal matrix that is independent of the state $s$. This is motivated by the normalized advantage function in Gu et al. (2016a).

`MLP`. $\psi_w(s, a)$ is assumed to be a neural network in which we first encode the state $s$ with a hidden layer, and then concatenate it with the action $a$ and pass them into another hidden layer before the output.

Further, we denote by `Value` the typical value function baseline, which corresponds to setting $\psi_w(s, a) = 0$ in our case. For the variance parameters $\theta_2$ of the Gaussian policy, we use formula (19) for `Linear` and `Quadratic`, but (18) for `MLP` due to the difficulty of calculating the second order derivative $\nabla_{a,a}\psi_w(s, a)$ in MLP. All the results we report are averaged over three random seeds. See Appendix for implementation details.

## 5.1 Comparing the Variance of different Gradient Estimators

We start with comparing the variance of the gradient estimators with different control variates. Figure 1 shows the results on Walker2d-v1, when we take a fixed policy obtained by running the vanilla PPO for 2000 steps and evaluate the variance of the different gradient estimators under different sample size $n$. In order to obtain unbiased estimates of the variance, we estimate all the baseline functions using a hold-out dataset with a large sample size. We find that our methods, especially those using the `MLP` and `Quadratic` baselines, obtain significantly lower variance than the typical value function baseline methods. In our other experiments of policy optimization, we used the data from the current policy to estimate $\phi$, which introduces a small bias theoretically, but was found to perform well empirically (see Appendix 7.4 for more discussion on this issue).

## 5.2 Comparison with Q-prop using TRPO for policy optimization

Next we want to check whether our Stein control variate will improve the sample efficiency of policy gradient methods over existing control variate, e.g. Q-prop (Gu et al., 2016b). One major advantage of Q-prop is that it can leverage the off-policy data to estimate $q_w(s, a)$. Here we compare our methods with the original implementation of Q-prop which incorporate this feature for policy optimization. Because the best existing version of Q-prop is implemented with TRPO, we implement a variant of our method with TRPO for fair comparison. The results on Hopper-v1 and Walker2d-v1 are shown in Figure 2, where we find that all Stein control variates, even including `FitQ+Linear`

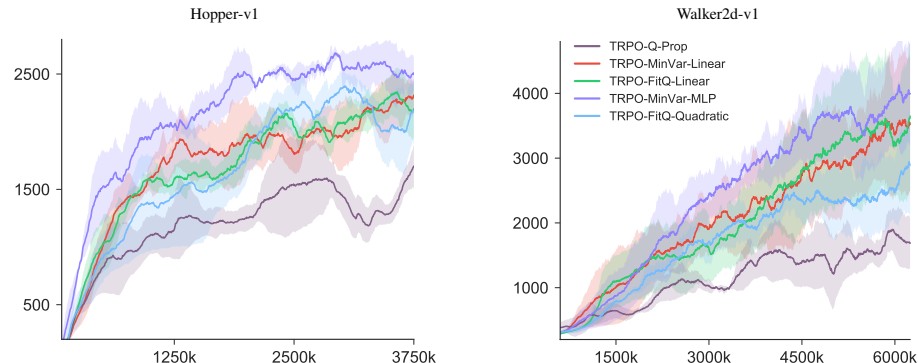

Figure 2: Evaluation of TRPO with Q-prop and Stein control variates on Hopper-v1 and Walker2d-v1.

| Function | Humanoid-v1 | | HumanoidStandup-v1 | |
|---|---|---|---|---|
| | MinVar | FitQ | MinVar | FitQ |
| MLP | **3847 ± 249.3** | 3334 ± 695.7 | **143314 ± 9471** | 139315 ± 10527 |
| Quadratic | 2356 ± 294.7 | **3563 ± 235.1** | 117962 ± 5798 | **141692 ± 3489** |
| Linear | 2547 ± 701.8 | 3404 ± 813.1 | 129393 ± 18574 | 132112 ± 11450 |
| Value | 2207 ± 554 | | 128765 ± 13440 | |

Table 1: Results of different control variates and methods for optimizing $\phi$, when combined with PPO. The reported results are the average reward at the 10000k-th time step on Humanoid-v1 and the 5000k-th time step on HumanoidStandup-v1.

and `MinVar+Linear`, outperform Q-prop on both tasks. This is somewhat surprising because the Q-prop compared here utilizes both on-policy and off-policy data to update $w$, while our methods use only on-policy data. We expect that we can further boost the performance by leveraging the off-policy data properly, which we leave it for future work. In addition, we noticed that the `Quadratic` baseline generally does not perform as well as it promises in Figure 1; this is probably because that in the setting of policy training we optimize $\phi_w$ for less number of iterations than what we do for evaluating a fixed policy in Figure 1, and it seems that `Quadratic` requires more iterations than `MLP` to converge well in practice.

## 5.3 PPO WITH DIFFERENT CONTROL VARIATES

Finally, we evaluate the different Stein control variates with the more recent proximal policy optimization (PPO) method which generally outperforms TRPO. We first test all the three types of $\phi$ listed above on Humanoid-v1 and HumanoidStandup-v1, and present the results in Table 1. We can see that all the three types of Stein control variates consistently outperform the value function baseline, and `Quadratic` and `MLP` tend to outperform `Linear` in general.

We further evaluate our methods on a more extensive list of tasks shown in Figure 3, where we only show the result of `PPO+MinVar+MLP` and `PPO+FitQ+MLP` which we find tend to perform the best according to Table 1. We can see that our methods can significantly outperform `PPO+Value` which is the vanilla PPO with the typical value function baseline (Heess et al., 2017).

It seems that `MinVar` tends to work better with `MLP` while `FitQ` works better with `Quadratic` in our settings. In general, we find that `MinVar+MLP` tends to perform the best in most cases. Note that the `MinVar` here is based on minimizing the approximate objective (27) specific to Gaussian policy, and it is possible that we can further improve the performance by directly minimizing the exact objective in (15) if an efficient implementation is made possible. We leave this a future direction.

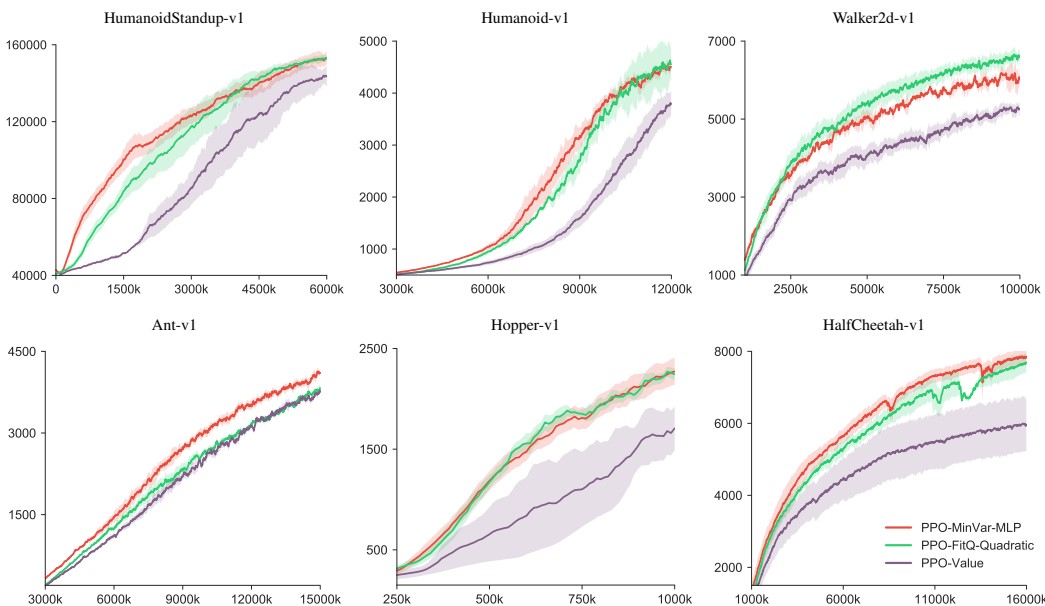

Figure 3: Evaluation of PPO with the value function baseline and Stein control variates across different Mujoco environments: HumanoidStandup-v1, Humanoid-v1, Walker2d-v1, Ant-v1 and Hopper-v1, HalfCheetah-v1.

## 6 CONCLUSION

We developed the Stein control variate, a new and general variance reduction method for obtaining sample efficiency in policy gradient methods. Our method generalizes several previous approaches. We demonstrated its practical advantages over existing methods, including Q-prop and value-function control variate, in several challenging RL tasks. In the future, we will investigate how to further boost the performance by utilizing the off-policy data, and search for more efficient ways to optimize $\phi$. We would also like to point out that our method can be useful in other challenging optimization tasks such as variational inference and Bayesian optimization where gradient estimation from noisy data remains a major challenge.

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

# 7 APPENDIX

## 7.1 PROOF OF THEOREM 3.1

*Proof.* Assume $a = f_\theta(s, \xi) + \xi_0$ where $\xi_0$ is Gaussian noise $\mathcal{N}(0, h^2)$ with a small variance $h$ that we will take $h \to 0^+$. Denote by $\pi(a, \xi | s)$ the joint density function of $(a, \xi)$ conditioned on $s$. It can be written as

$$\pi(a, \xi | s) = \pi(a | s, \xi) \pi(\xi) \propto \exp\left( -\frac{1}{2h^2} ||a - f_\theta(s, \xi)||^2 \right) \pi(\xi).$$

Taking the derivative of $\log \pi(a, \xi | s)$ w.r.t. $a$ gives

$$\nabla_a \log \pi(a, \xi | s) = -\frac{1}{h^2} (a - f_\theta(s, \xi)).$$

Similarly, taking the derivative of $\log \pi(a, \xi | s)$ w.r.t. $\theta$, we have

$$\nabla_\theta \log \pi(a, \xi | s) = \frac{1}{h^2} \nabla_\theta f_\theta(s, \xi) (a - f_\theta(s, \xi))$$
$$= -\nabla_\theta f_\theta(s, \xi) \nabla_a \log \pi(a, \xi | s).$$

Multiplying both sides with $\phi(s, a)$ and taking the conditional expectation yield

$$\mathbb{E}_{\pi(a, \xi | s)}\big[ \nabla_\theta \log \pi(a, \xi | s) \phi(s, a) \big] = -\mathbb{E}_{\pi(a, \xi | s)}\big[ \nabla_\theta f_\theta(s, \xi) \nabla_a \log \pi(a, \xi | s) \phi(s, a) \big]$$
$$= \mathbb{E}_{\pi(\xi)}\big[ \nabla_\theta f_\theta(s, \xi) \mathbb{E}_{\pi(a | s, \xi)}\big[ -\nabla_a \log \pi(a, \xi | s) \phi(s, a) \big] \big]$$
$$= \mathbb{E}_{\pi(\xi)}\big[ \nabla_\theta f_\theta(s, \xi) \mathbb{E}_{\pi(a | s, \xi)}\big[ \nabla_a \phi(s, a) \big] \big]$$
$$= \mathbb{E}_{\pi(a, \xi | s)}\big[ \nabla_\theta f_\theta(s, \xi) \nabla_a \phi(s, a) \big] \tag{21}$$

where the third equality comes from Stein's identity (5) of $\pi(a | \xi, s)$. One the other hand,

$$\mathbb{E}_{\pi(a, \xi | s)}\big[ \nabla_\theta \log \pi(a, \xi | s) \phi(s, a) \big]$$
$$= \mathbb{E}_{\pi(a, \xi | s)}\big[ \nabla_\theta \log \pi(a | s) \phi(s, a) \big] + \mathbb{E}_{\pi(a, \xi | s)}\big[ \nabla_\theta \log \pi(\xi | s, a) \phi(s, a) \big] \tag{22}$$
$$= \mathbb{E}_{\pi(a, \xi | s)}\big[ \nabla_\theta \log \pi(a | s) \phi(s, a) \big], \tag{23}$$

where the second term of (22) equals zero because

$$\mathbb{E}_{\pi(a, \xi | s)}\big[ \nabla_\theta \log \pi(\xi | s, a) \phi(s, a) \big] = \mathbb{E}_{\pi(a | s)}\big[ \mathbb{E}_{\pi(\xi | s, a)}\big[ \nabla_\theta \log \pi(\xi | s, a) \big] \big] \phi(s, a) \big] = 0.$$

Combining (21) and (23) gives the result:

$$\mathbb{E}_{\pi(a | s)}\big[ \nabla_\theta \log \pi(a | s) \phi(s, a) \big] = \mathbb{E}_{\pi(a, \xi | s)}\big[ \nabla_\theta f_\theta(s, \xi) \nabla_a \phi(s, a) \big]$$

The above result does not depend on $h$, and hence holds when $h \to 0^+$. This completes the proof. □

## 7.2 ESTIMATING $\phi$ FOR GAUSSIAN POLICIES

The parameters $w$ in $\phi$ should be ideally estimated by minimizing the variance of the gradient estimator $\mathrm{var}(\nabla_\theta J(\theta))$ using (15). Unfortunately, it is computationally slow and memory inefficient to directly solve (15) with the current deep learning platforms, due to the limitation of the auto-differentiation implementations. In general, this problem might be solved with a customized implementation of gradient calculation as a future work. But in the case of Gaussian policies, we find minimizing $\mathrm{var}(\hat{\nabla}_\mu J(\theta)) + \mathrm{var}(\hat{\nabla}_\Sigma J(\theta))$ provides an approximation that we find works in our experiments.

More specifically, recall that Gaussian policy has a form of

$$\pi(a \mid s) \propto \frac{1}{\sqrt{|\Sigma(s)|}} \exp\left( -\frac{1}{2} (a - \mu(s))^\top \Sigma(s)^{-1} (a - \mu(s)) \right),$$

where $\mu(s)$ and $\Sigma(s)$ are parametric functions of state s, and $|\Sigma|$ is the determinant of $\Sigma$. Following Eq (8) we have

$$\nabla_\mu J(\theta) = \mathbb{E}_\pi \left[ -\nabla_a \log \pi(a | s)(Q^\pi(s, a) - \phi(s, a)) + \nabla_a \phi(s, a) \right], \tag{24}$$

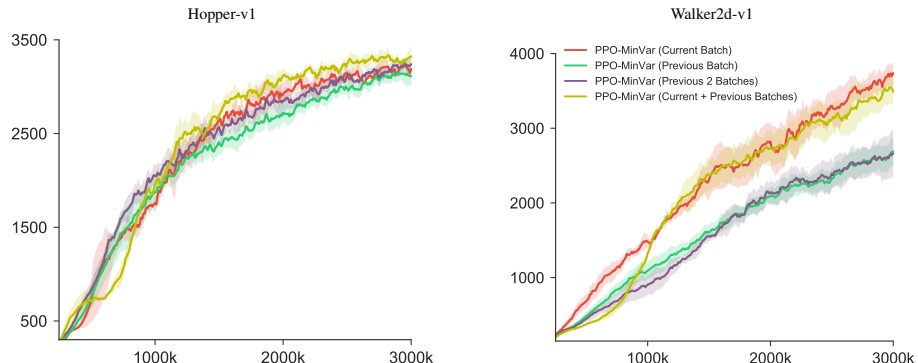

Figure 4: Evaluation of PPO with Stein control variate when $\phi$ is estimated based on data from different iterations. The architecture of $\phi$ is choosen to be `MLP`.

where we use the fact that $\nabla_\mu f(s, \xi) = 1$ and

$$\nabla_\mu \log \pi(a|s) = -\nabla_a \log \pi(a|s) = \Sigma(s)^{-1}(a - \mu(s)).$$

Similarly, following Eq (18), we have

$$\nabla_\Sigma J(\theta) = \mathbb{E}_\pi \left[ \nabla_\Sigma \log \pi(a|s)(Q^\pi(s,a) - \phi(s,a)) - \frac{1}{2}\nabla_a \log \pi(a|s)\nabla_a \phi(s,a)^\top \right] \quad (25)$$

where

$$\nabla_\Sigma \log \pi(a|s) = \frac{1}{2}\left(-\Sigma^{-1}(s) + \Sigma(s)^{-1}(a - \mu(s))(a - \mu(s))^\top \Sigma(s)^{-1}\right).$$

And Eq (19) reduces to

$$\nabla_\Sigma J(\theta) = \mathbb{E}_\pi \left[ \nabla_\Sigma \log \pi(a|s)(Q^\pi(s,a) - \phi(s,a)) + \frac{1}{2}\nabla_{a,a}\phi(s,a) \right]. \quad (26)$$

Because the baseline function does not change the expectations in (24) and (25), we can frame $\min_w \text{var}(\hat{\nabla}_\mu J(\theta)) + \text{var}(\hat{\nabla}_\Sigma J(\theta))$ into

$$\min_w \sum_{t=1}^n \|g_\mu(s_t, a_t)\|_2^2 + \|g_\Sigma(s_t, a_t)\|_F^2, \quad (27)$$

where $g_\mu$ and $g_\Sigma$ are the integrands in (24) and (25) (or (26)) respectively, that is, $g_\mu(s,a) = -\nabla_a \log \pi(a|s)(Q^\pi(s,a) - \phi(s,a)) + \nabla_a \phi(s,a)$ and $g_\Sigma(s,a) = \nabla_\Sigma \log \pi(a|s)(Q^\pi(s,a) - \phi(s,a)) - \frac{1}{2}\nabla_a \log \pi(a|s)\nabla_a \phi(s,a)^\top$. Here $\|A\|_F^2 := \sum_{ij} A_{ij}^2$ is the matrix Frobenius norm.

### 7.3 EXPERIMENT DETAILS

The advantage estimation $\hat{A}^\pi(s_t, a_t)$ in Eq 16 is done by GAE with $\lambda = 0.98$, and $\gamma = 0.995$ (Schulman et al., 2016), and correspondingly, $\hat{Q}^\pi(s_t, a_t) = \hat{A}^\pi(s_t, a_t) + \hat{V}^\pi(s_t)$ in (9). Observations and advantage are normalized as suggested by Heess et al. (2017). The neural networks of the policies $\pi(a|s)$ and baseline functions $\phi_w(s, a)$ use *Relu* activation units, and the neural network of the value function $\hat{V}^\pi(s)$ uses *Tanh* activation units. All our results use Gaussian MLP policy in our experiments with a neural-network mean and a constant diagonal covariance matrix.

Denote by $d_s$ and $d_a$ the dimension of the states $s$ and action $a$, respectively. Network sizes are follows: On Humanoid-v1 and HuamnoidStandup-v1, we use $(d_s, \sqrt{d_a \cdot 5}, 5)$ for both policy network and value network; On other Mujoco environments, we use $(10 \cdot d_s, \sqrt{10 \cdot d_s \cdot 5}, 5)$ for both policy network and value network, with learning rate $\frac{0.0009}{\sqrt{(d_s \cdot 5)}}$ for policy network and $\frac{0.0001}{\sqrt{(d_s \cdot 5)}}$ for value network. The network for $\phi$ is $(100, 100)$ with state as the input and the action concatenated with the second layer.

All experiments of PPO with Stein control variate selects the best learning rate from {0.001, 0.0005, 0.0001} for $\phi$ networks. We use ADAM (Kingma & Ba, 2014) for gradient descent and evaluate the policy every 20 iterations. Stein control variate is trained for the best iteration in range of {250, 300, 400, 500, 800}.

## 7.4 ESTIMATING $\phi$ USING DATA FROM PREVIOUS ITERATIONS

Our experiments on policy optimization estimate $\phi$ based on the data from the current iteration. Although this theoretically introduces a bias into the gradient estimator, we find it works well empirically in our experiments. In order to exam the effect of such bias, we tested a variant of PPO-MinVar-MLP which fits $\phi$ using data from the previous iteration, or previous two iterations, both of which do not introduce additional bias due to the dependency of $\phi$ on the data. Figure 4 shows the results in Hopper-v1 and Walker2d-v1, where we find that using the data from previous iterations does not seem to improve the result. The may be because during the policy optimization, the updates of $\phi$ are early stopped and hence do not introduce overfitting even when it is based on the data from the current iteration.

