# OpenReview forum: "Action-dependent Control Variates for Policy Optimization via Stein Identity"
_ICLR.cc/2018/Conference — Accept (Poster)_

### Official Review · AnonReviewer1 · 2017-11-26
**Nice work; some suggestions in order to improve the paper are provided.**

**Rating:** 7
**Confidence:** 3

**Review:**

In this work, the authors suggest the use of control variate schemes for estimating gradient values, within a reinforcement learning  framework. The authors also introduce a specific control variate technique based on the so-called Stein’s identity. The paper is interesting and well-written.

I have some question and some consideration that can be useful for improving the appealing of the paper.

- I believe that different Monte Carlo (or Quasi-Monte Carlo) strategies can be applied in order to estimate the integral (expected value) in Eq. (1), as also suggested in this work. Are there other alternatives in the literature? Please, please discuss and cite some papers if required.

- I suggest to divide Section 3.1 in two subsections. The first one introducing Stein’s identity and the related comments that you need, and a second one, starting after Theorem 3.1, with title “Stein Control Variate”.

-  Please also discuss the relationships, connections, and possible applications of your technique to other algorithms used in Bayesian optimization, active learning and/or sequential learning, for instance as

M. U. Gutmann and J. Corander, “Bayesian optimization for likelihood-free inference of simulator-based statistical mod- els,” Journal of Machine Learning Research, vol. 16, pp. 4256– 4302, 2015.

G. da Silva Ferreira and D. Gamerman, “Optimal design in geostatistics under preferential sampling,” Bayesian Analysis, vol. 10, no. 3, pp. 711–735, 2015.

L. Martino, J. Vicent, G. Camps-Valls, "Automatic Emulator and Optimized Look-up Table Generation for Radiative Transfer Models", IEEE International Geoscience and Remote Sensing Symposium (IGARSS), 2017.

-  Please also discuss the dependence of your algorithm with respect to the starting baseline function \phi_0.

---

> ### Author Response · Authors · 2018-01-02
> **Rebuttal to AnonReview3:**
>
> Thank you very much for the thoughtful feedbacks, with which we could further improve our paper.
>
> * Different Mote Carlo strategies for estimating the integral? Because the setting here is model-free, that is, we only have a black-box to simulate from the environment, without knowing the underlying distribution, there more limited MC strategies can be used than typical integration problems. Nevertheless, some advanced techniques such as Bayesian quadrature can be used (Ghavamzadeh et al. Bayesian Policy Gradient and Actor-Critic Algorithms).
>
> * We will modify Section 3.1 according to your suggestion.
>
> *  It would be very interesting to consider the application of this technique to Bayesian optimization. We will certainly discuss the possibility in the future work section.

---

### Official Review · AnonReviewer3 · 2017-11-26
**Nice Application of Stein's identity**

**Rating:** 7
**Confidence:** 3

**Review:**

This paper proposed a class of control variate methods based on Stein's identity. Stein's identity has been widely used in classical statistics and recently in statistical machine learning literature. Nevertheless, applying Stein's identity to estimating policy gradient is a novel approach in reinforcement learning community. To me, this approach is the right way of constructing control variates for estimating policy gradient. The authors also did a good job in connecting with existing works and gave concrete examples for Gaussian policies. The experimental results also look promising.

It would be nice to include some theoretical analyses like under what conditions, the proposed method can achieve smaller sample complexity than existing works.

Overall this is a strong paper and I recommend to accept.

---

> ### Author Response · Authors · 2018-01-02
> **Rebuttal to AnonReview2:**
>
> Thank you very much for the review. We are interested in studying theoretical properties of the estimators as well, but because of the non-convex nature of RL problems, it may be better to start theoretical analysis in simpler cases such as convex problems, in which some interesting results on convergence rate can be potentially be obtained (perhaps in connection to stochastic variance reduced gradient in some way).

---

### Official Review · AnonReviewer2 · 2017-11-28
**Good empirical study of modifications to action-dependent baselines**

**Rating:** 7
**Confidence:** 4

**Review:**

The paper proposes action-dependent baselines for reducing variance in policy gradient, through the derivation based on Stein’s identity and control functionals. The method relates closely to prior work on action-dependent baselines, but explores in particular on-policy fitting and a few other design choices that empirically improve the performance.

A criticism of the paper is that it does not require Stein’s identity/control functionals literature to derive Eq. 8, since it can be derived similarly to linear control variate and it has also previously been discussed in IPG [Gu et. al., 2017] as reparameterizable control variate. The derivation through Stein’s identity does not seem to provide additional insights/algorithm designs beyond direct derivation through reparameterization trick.

The empirical results appear promising, and in particular in comparison with Q-Prop, which fits Q-function using off-policy TD learning. However, the discussion on the causes of the difference should be elaborated much more, as it appears there are substantial differences besides on-policy/off-policy fitting of the Q, such as:

-FitLinear fits linear Q (through parameterization based on linearization of Q) using on-policy learning, rather than fitting nonlinear Q and then at application time linearize around the mean action. A closer comparison would be to use same locally linear Q function for off-policy learning in Q-Prop.

-The use of on-policy fitted value baseline within Q-function parameterization during on-policy fitting is nice. Similar comparison should be done with off-policy fitting in Q-Prop.

I wonder if on-policy fitting of Q can be elaborated more. Specifically, on-policy fitting of V seems to require a few design details to have best performance [GAE, Schulman et. al., 2016]: fitting on previous batch instead of current batch to avoid overfitting  (this is expected for your method as well, since by fitting to current batch the control variate then depends nontrivially on samples that are being applied), and possible use of trust-region regularization to prevent V from changing too much across iterations.

The paper presents promising results with direct on-policy fitting of action-dependent baseline, which is promising since it does not require long training iterations as in off-policy fitting in Q-Prop. As discussed above, it is encouraged to elaborate other potential causes that led to performance differences. The experimental results are presented well for a range of Mujoco tasks.

Pros:

-Simple, effective method that appears readily available to be incorporated to any on-policy PG methods without significantly increase in computational time

-Good empirical evaluation

Cons:

-The name Stein control variate seems misleading since the algorithm/method does not rely on derivation through Stein’s identity etc. and does not inherit novel insights due to this derivation.

---

> ### Author Response · Authors · 2018-01-02
> **Rebuttal to AnonReview1:**
>
> Thank you very much for the review and pointing out potential improvements. The followings are the response to your comments:
>
> * Thanks for pointing out IPG and on-policy vs. off-policy fitting; we will provide a thorough discussion on this. We have been mainly focussing on fitting \phi with on-policy, because the optimal control variates should theoretically depend on the current policy and hence "on-policy" in its nature. However, we did experiment ways to use additional off-policy data to our update and find that using additional off-policy data can in fact further improve our method. We find it is hard to have a fair comparison between on policy vs. off-policy fitting because it largely depends on how we implement each of them. Instead, an interesting future direction for us is to investigate principled ways to combine them to improve beyond what we can achieve now.
>
> We should point out the difference between IPG and our method is not only the way we fit \phi, another perhaps more significant difference is that IPG (depending which particular version) also averages over off-policy data when estimating the gradient, while our method always only averages over the on-policy data.
>
> * In our comparison, Q-prop also uses an on-policy fitted value function inside the Q-function.
>
> * Thank you very much for suggesting better ways of on-policy fitting of V. We are interested in testing them for future works. Currently, V is fitted by all the current data which theoretically introduces a (possibly small) bias because the current data is used twice in the gradient estimator, so using the data from the previous iteration may yield improvement.
>
>
> * Regarding the name, although it turned out our result can be derived using reparameterization trick, Stein's identity is what motivated this work originally, and we lean towards keeping it as the motivation since Stein's identity generally provides a principled way to think about control variates (which essentially requires zero-expectation identities mathematically).
>
> Stein's identity and reparameterization trick are two orthogonal ways to think about this work, and it is useful to keep both of them to give a more comprehensive view.  It is not true that Stein's identity is not directly useful in our work: By using (the original) Stein's identity on the top of the basic formula, we can derive a different control variate for Gaussian policy that has lower variance (and it is what we used in experiments). It is possible that we can further generalize the result by using Stein's identity in more creative ways. On the other hand, we will emphasize more the role of reparameterization trick in the revision.

---

> > ### Comment · AnonReviewer2 · 2018-01-11
> > **Thank you for the rebuttal; I will keep my original score.**
> >
> > Good to see you included additional discussions. Note that if the second term is estimated zero-variance per state (e.g. sampling many actions instead of using single-sample reparameterized gradient, or pick a control variate that you can integrate directly under the policy), the optimal control variate is Q^\pi, which can be learned using any policy evaluation technique, on-policy or off-policy; it's discussed in Q-prop as well.
> >
> > Re: using the same samples to fit the control variate (many gradient updates) and apply to themselves. This could introduce non-trivial bias. It's easy to imagine that in such case, the first term can go to zero, because assuming finite, diverse enough samples, Q can potentially fit all sample returns. In such cases, it's important not only to report variance as you have estimated, but also bias. It's highly encouraged to discuss/include a few more details on this in the final paper.

---

### Public Comment · (anonymous) · 2017-11-07
**Comment**

I'm wondering if in fact what is suggested as Stein Control Varite is not indeed similar (if not the same) with the technique proposed here: https://arxiv.org/abs/1711.00123 ?

---

> ### Author Response · Authors · 2017-11-09
> **Thank you for comment**
>
> Thank you for pointing us to this independent ICLR submission. It is highly relevant. Their estimator in the RL setting (their Eq 11) is mathematically equivalent to ours. However, our paper is derived from a different perspective and give more comprehensive results on reinforcement learning specifically.
>
> 1) Our work focuses on RL. By combining with PPO and TRPO, we obtain significant improvement on challenging RL tasks such as Humanoid-v1 and HumanoidStandup-v1. We also proposed and tested different architectures and optimization methods for the control variates, providing guidance on what may work best in practice. We explicitly establish the connection with  Q-prop(Gu et al., 2016b), which can be viewed as our method with linear control variates.
>
> 2)  Our work was motivated by Stein’s identity and control functionals (Oates et al. 2017), and hence develops a connection between Stein’s identity and reparameterization trick which can be itself useful. For example, for Gaussian policy, we further derive a different update rule with lower variance by utilizing Stein’s Identity twice.

---

> > ### Public Comment · (anonymous) · 2017-11-10
> > **Good to have a discussion**
> >
> > So, just to emphasise the similarity, and note that since these are parallel submissions by no means I intend to diminish your contributions, just I think the connection is interesting. Equation (8) from your paper is exactly equivalent to Equation (6) (LAX estimator) of the paper, specifically by setting:
> > pi(a|theta) = p(b|theta)
> > Q(s,a) = f(b)
> > phi(s,a) = c_phi(b)
> > f(s, eps|theta) = T(eps, theta)
> > Similar to your remark in Equation (13) the other authors just below their Equation (6) mention that taking the "control" function equal the original one (if that is differentiable) we recover the path gradient. Additionally, they also suggest optimizing the "control" function by minimizing the variance.
> >
> > Regarding, equation (18) and the Gaussian policy indeed it is an interesting observation that we can apply this a second time and get a potentially lower variance estimator. This in fact I think is a more general result that if the derivatives depend on epsilon you can reapply the procedure, but don't cite me on that. Potentially, investigating/generalizing that might be interesting.

---

### Public Comment · (anonymous) · 2017-12-03
**Videos and code**

Great results and very interesting paper!

Do you plan to share video some of the learnt policies? And do you plan to share a code later?

---

> ### Author Response · Authors · 2017-12-06
> **Re:Videos and code**
>
> Hi, thanks for your interest, code has released here: https://github.com/DartML/PPO-Stein-Control-Variate.
>
> We plan to share the videos of learned policies.

---

> > ### Public Comment · (anonymous) · 2017-12-06
> > **Thanks**
> >
> > Thanks a lot!

---

### Author Response · Authors · 2018-01-06
**Changes made to the paper**

Dear Reviewers,

We just submitted a modification of the paper. The main changes are

1. Modified the title.

2. We split Section 3.1 following the suggestion of AnonReviewer1.

3. We cited and discussed IPG.

4. The original code that generates figure 1 had a problem when calculating the variance of the gradient estimator. We fixed it and updated figure 1.

5. In our policy optimization method, we estimate phi based on the data from the current iteration. This introduces a (typically negotiable) bias because the data were used for twice. A way to avoid this is to estimate phi based on data from the previous iterations. We studied the effect of this bias and empirically find that using the previous data, although eliminates this bias,  does not seem to improve the performance (see Appendix 7.3 and more discussion), and our current version tends to perform better. We clarified this point in the text as well.

We will further improve the paper based on the reviewers' comments.

---

### Public Comment · (anonymous) · 2018-01-13
**Discrepancies in Fig 1**

After several exchanges with the authors, we have been unable to replicate the results produced in Figure 1 that show the improvement of an action-dependent control variate. As the authors note, several bugs have affected Figure 1. Using the latest code provided by the authors, we do not find a reduction in variance with a state-action control variate compared to a state-only control variate.

---

> ### Author Response · Authors · 2018-02-14
> **Response**
>
> Thanks for your comments,
>
> Figure 1 is in fact reproducible since the commenter is also able to repeat it. What he mentioned is that if we add an additional state only control variate, the performance is close to an additoinal action-dependent control variate. In the current Mujoco experiments, we agree that we can not find clear improvement of action dependent control variates over state-only control variates, although we did find improvement on a toy MDP example (not for Mujoco experiments).

---

### Decision · Program_Chairs · 2018-01-29
**ICLR 2018 Conference Acceptance Decision**

**Decision:**

Accept (Poster)

**Comment:**

Thank you for submitting you paper to ICLR. The reviewers agree that the paper’s development of action-dependent baselines for reducing variance in policy gradient is a strong contribution and that the use of Stein's identity to provide a principled way to think about control variates is sensible. The revision clarified an number of the reviewers’ questions and the resulting paper is suitable for publication in ICLR.